# Effects of an IgE receptor polymorphism acting on immunity, susceptibility to infection, and reproduction in a wild rodent

**Klara M Wanelik[1]\*, Mike Begon[1], Janette E Bradley[2], Ida M Friberg[3], Joseph A Jackson[3], Christopher H Taylor[2], Steve Paterson[1]**

[1]Institute of Infection, Veterinary and Ecological Sciences, University of Liverpool, Liverpool, United Kingdom; [2]School of Life Sciences, University of Nottingham, Nottingham, United Kingdom; [3]School of Environment and Life Sciences, University of Salford, Salford, United Kingdom

**\*For correspondence:**
klara.wanelik@biology.ox.ac.uk

**Competing interest:** The authors declare that no competing interests exist.

**Abstract** The genotype of an individual is an important predictor of their immune function, and subsequently, their ability to control or avoid infection and ultimately contribute offspring to the next generation. However, the same genotype, subjected to different intrinsic and/or extrinsic environments, can also result in different phenotypic outcomes, which can be missed in controlled laboratory studies. Natural wildlife populations, which capture both genotypic and environmental variability, provide an opportunity to more fully understand the phenotypic expression of genetic variation. We identified a synonymous polymorphism in the high-affinity Immunoglobulin E (IgE) receptor (GC and non-GC haplotypes) that has sex-dependent effects on immune gene expression, susceptibility to infection, and reproductive success of individuals in a natural population of field voles (*Microtus agrestis*). We found that the effect of the GC haplotype on the expression of immune genes differed between sexes. Regardless of sex, both pro-inflammatory and anti-inflammatory genes were more highly relatively expressed in individuals with the GC haplotype than individuals without the haplotype. However, males with the GC haplotype showed a stronger signal for pro-inflammatory genes, while females showed a stronger signal for anti-inflammatory genes. Furthermore, we found an effect of the GC haplotype on the probability of infection with a common microparasite, *Babesia microti*, in females – with females carrying the GC haplotype being more likely to be infected. Finally, we found an effect of the GC haplotype on reproductive success in males – with males carrying the GC haplotype having a lower reproductive success. This is a rare example of a polymorphism whose consequences we are able to follow across immunity, infection, and reproduction for both males and females in a natural population.

## Editor's evaluation

This study provides an unusually comprehensive analysis of the associations between polymorphism in an immune gene, the immunoglobulin E receptor Fcer1a, and immune responses, resistance to infection, and reproductive fitness in a wild rodent population. The investigators find that these effects appear to be sex-specific. This study provides critical observations of the possible consequences of immune polymorphisms in wild populations and should be of interest to immunologists, evolutionary biologists, and ecologists investigating genotype-phenotype relationships and potential life-history tradeoffs.

## Introduction

In order for an individual to control or avoid infection and ultimately contribute offspring to the next generation, they must have a well-functioning immune system (*Møller and Saino, 2004*). An individual's immune function is in part determined by their genotype (e.g., *Paterson et al., 1998*; *Wanelik et al., 2018*; *Turner et al., 2011*). Individuals within a population differ from each other in the genotypes they carry, with the potential for these different genotypes to affect the activity of different immune components, and thereby influence susceptibility to infection and reproduction and survivorship. Indeed, variability in immune responses to infection is well-characterized. The phenotypic expression of a genotype is dependent on the environment, including the sex of an individual (intrinsic environment) and exposure to infection (extrinsic environment). Since capturing genotypic and environmental variability in controlled laboratory studies is problematic, natural wildlife populations have been proposed as models in which to understand the phenotypic expression of genetic variation and its consequences for susceptibility to infection and reproduction.

Studies of natural populations have explored the effects of genotype on immune phenotype and have observed consequences for susceptibility to infection. Most notably, variability in the genes of the major histocompatibility complex (MHC) has been associated with resistance to intestinal nematodes in domestic sheep (*Paterson et al., 1998*) and with resistance to malaria, hepatitis, and AIDS in humans (*Hill et al., 1991*; *Carrington et al., 1999*; *Thursz et al., 1997*). The role of variability elsewhere in the genome, for shaping immune phenotype, has also been studied (*Wanelik et al., 2018*; *Turner et al., 2011*). However, it remains challenging to follow the consequences of a genotypic effect for immunity, infection, and reproduction and to account for any sex-dependent expression of a genotype. This is because of the difficulty in obtaining phenotypic data across immune, infection, and reproductive traits, especially for large enough sample sizes to test for data-hungry genotype by sex interactions (*Ober et al., 2008*). In many cases, sex and other environmental factors are considered as a confounding variable to be controlled for in order not to hide any subtle genetic associations (*Paterson et al., 1998*). Other studies focus on a single sex for the sake of simplicity (*Jackson et al., 2014*). More recently, however, there has been a growing body of large-scale field studies of natural populations able to apply genetic and immunological methods to follow large numbers of individuals, exposed to a challenging environment and with varying genetic backgrounds, throughout their lives. This allows us to make a more complete assessment of the impacts of genotype throughout the life of an individual, whether male or female. For example, *Graham et al., 2010* found evidence for heritable variation in immunity associated with sex-dependent effects on Soay sheep reproduction. However, we know of no documented example of a polymorphism affecting immunity, susceptibility to infection, and reproduction in a natural population investigated in males and females. Here, we use a wild rodent, the field vole (*Microtus agrestis*), as a model in which to do this. Wild rodents, in particular, offer an opportunity to quickly follow large numbers of individuals throughout their lives, given their short lifespans. They also offer the opportunity to draw on the immunological and genetics resources developed for laboratory rodents, while providing a much more realistic ecological model of human populations (*Turner et al., 2014*; *Turner and Paterson, 2013*).

Immunoglobulin E (IgE)-mediated responses are associated with defense against helminths (*Gounni et al., 1994*) and with allergy (*Tomassini et al., 1991*). They are controlled by the high-affinity IgE receptor, FCER1, which is found on the surface of various immune effector cells, for example, mast cells, basophils, and eosinophils (*Daeron and Nimmerjahn, 2014*). In humans, naturally occurring polymorphisms in FCER1 are known to affect an individual's serum IgE levels, with consequences for their susceptibility to infection (*Weidinger et al., 2008*; *Granada et al., 2012*) and their risk of developing inflammatory disease (*Hasegawa et al., 2003*; *Potaczek et al., 2006*; *Zhou et al., 2012*; *Niwa et al., 2010*). Furthermore, sex differences in serum IgE levels (*Weiss et al., 2006*) and the incidence of IgE-mediated inflammatory disease (*Chen et al., 2008*) have been documented in humans, suggesting that any polymorphism in this pathway is likely to experience different contexts in males and females. Indeed, one study found evidence for a polymorphism in the *Fcer1a* gene (the alpha chain of FCER1) whose association with susceptibility to systemic lupus erythematosus (a chronic inflammatory disease) differed between males and females (*Yang et al., 2013*).

In a previous study of a natural population of *M. agrestis*, we found that males carrying the GC haplotype of the *Fcer1a* gene expressed the transcription factor GATA3 at a lower level than males carrying non-GC haplotypes (*Wanelik et al., 2018*). GATA3 is a biomarker of tolerance to macroparasites in

mature males in our population (macroparasite infection gives rise to increased expression of GATA3, which gives rise to improved body condition and survival; *Jackson et al., 2014*). Here, we explore the effects of this GC haplotype further in both males and females to analyze the effect of genotype acting across immune gene expression, infection susceptibility and reproductive success.

## Results

We sampled a natural population of *M. agrestis* in Kielder Forest, Northumberland, UK, over 3 years (2015–2017) and across seven different sites. Our study involved a cross-sectional component (n = 317 destructively sampled voles) and a longitudinal component (n = 850 marked individuals monitored through time, with n = 2387 sampling points). We tested the consequences of the GC haplotype of the *Fcer1a* gene using both cross-sectional and longitudinal components of our study. As well as the GC haplotype (present at a frequency of 0.08), three other haplotypes were present in our study population: AC haplotype, AT haplotype, and GT haplotype, present at frequencies of 0.81, 0.10, and 0.01, respectively. The two single-nucleotide polymorphisms (SNPs) composing the haplotype were found to be tightly linked ($r^2$ = 0.50; D' = 0.70).

### The GC haplotype has effects on inflammation that differ between sexes

In humans, naturally occurring polymorphisms in the *Fcer1a* gene have previously been linked to inflammatory disease (*Hasegawa et al., 2003*; *Potaczek et al., 2006*; *Zhou et al., 2012*; *Niwa et al., 2010*). Therefore, we used the cross-sectional component of our study to test the effects of the GC haplotype on inflammation in males and females. Differential gene expression (DGE) analysis was performed on unstimulated splenocytes taken from 53 males and 31 females assayed by RNASeq, with the aim of identifying individual genes that were differentially expressed between those individuals with and without a copy of the GC haplotype. This DGE analysis showed that the identity of top differentially expressed genes differed between the sexes. In males, the top differentially expressed immune gene was the cytokine, *Il33* (log fold change [logFC] = 2.76, p<0.001, *q* < 0.001; *Appendix 1—table 2*) while in females it was the suppressor of cytokine signaling *Socs3* (logFC = 1.07, p<0.001, *q* = 0.05; *Appendix 1—table 3*). Looking at the ranking of each top differentially expressed immune gene in the opposite sex strengthens the case for differing effects in males and females, with *Il33* ranked markedly lower in females (rank = 8224/12904, logFC = 0.29, p=0.70, *q* = 1.00) and *Socs3* markedly lower in males (rank = 10886/12904, logFC = –0.05, p=0.84, *q* = 1.00). *Il33* is commonly associated with the anti-helminthic response (*Liew et al., 2010*) and *Socs3* with regulation of the immune response more broadly (*Carow and Rottenberg, 2014*). Given the link between *Il33* and the antihelminthic response (and more generally, IgE-mediated responses and the antihelminthic response), we repeated the DGE analysis while controlling for cestode burden, but this had little effect on our results (same top differentially expressed immune genes; see *Appendix 1—table 4* and *Appendix 1—table 5*), suggesting that these effects were not driven by differences in cestode infection.

Both *Il33* and *Socs3* also share an association with the inflammatory response (*Carow and Rottenberg, 2014*; *Cayrol and Girard, 2014*). While *Il33* positively regulates this response (appearing in the gene set GO:0050729), *Socs3* negatively regulates it (GO:0050728). To test whether these effects on the inflammatory response were limited to these genes or were more widespread, we performed a gene set enrichment analysis (GSEA) that looked at the rankings of all genes present in each of these gene sets (GO:0050729 and GO:0050728). This analysis showed that both gene sets were more highly relatively expressed in individuals with the GC haplotype than individuals without the haplotype, and that this was true for both males and females. However, males with the GC haplotype showed a stronger signal for genes that positively regulate the inflammatory response (GO:0050729: p=0.007; GO:0050728: p=0.04; *Figure 1A*, upper panel) while females with the GC haplotype showed a stronger signal for genes that negatively regulate the inflammatory response (GO:0050728: p=0.001; GO:0050728: p=0.04; *Figure 1A*, lower panel).

To further explore these effects on the inflammatory response, we used an independent dataset for males and females whose spleens were stimulated with two immune agonists and assayed by Q-PCR (for a panel of 18 immune genes with limited redundancy; see 'Materials and methods' for how these genes were selected). From this ex vivo assay, one can gain insight into the types of immune response

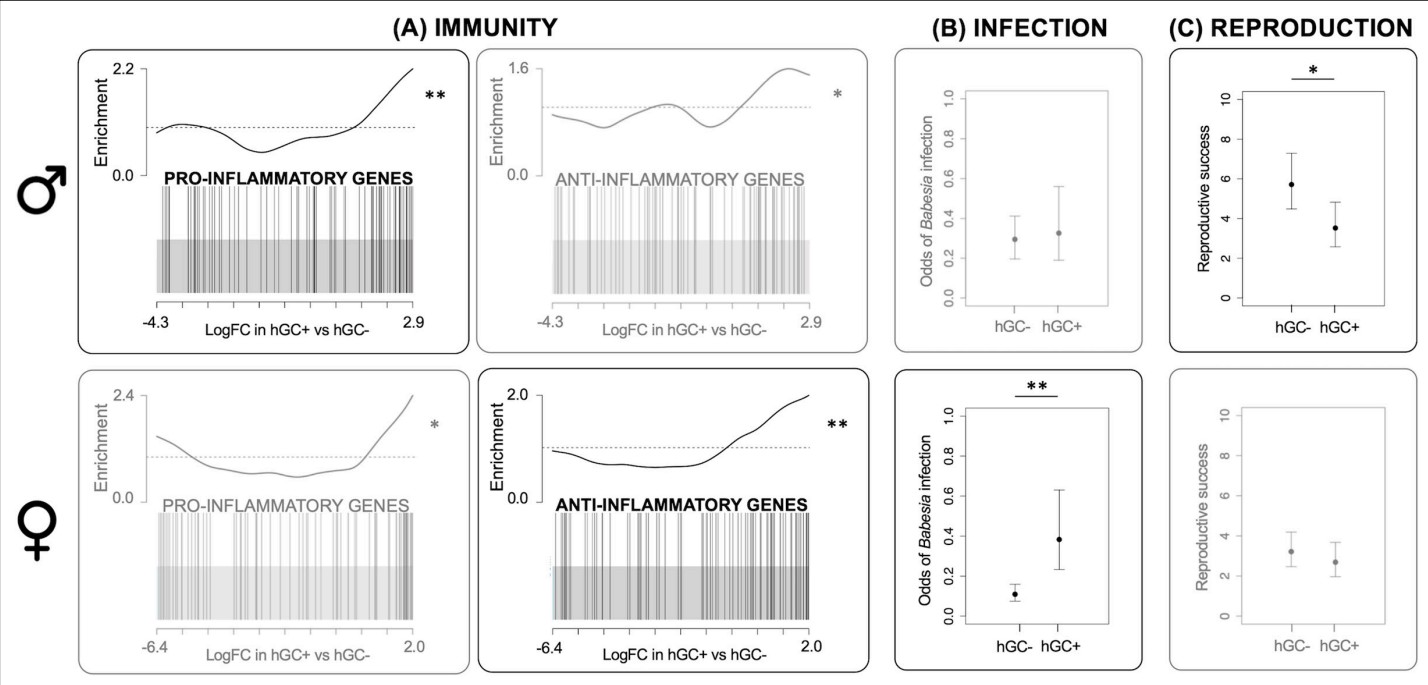

**Figure 1.** Effects of GC haplotype (hGC). Upper panel: males. Lower panel: females: (**A**) Unstimulated immune gene expression: Barcode plots showing enrichment of the GO terms GO:0050729 (pro-inflammatory genes) and GO:0050728 (anti-inflammatory genes) in unstimulated splenocytes taken from individuals with (hGC+) vs. without (hGC-) the haplotype, showing that males with the haplotype have a pro-inflammatory bias, whereas females have an anti-inflammatory bias. In each plot, x-axis shows log fold change (logFC) in hGC+ vs. hGC-, black bars represent genes annotated with the GO terms and the worm shows relative enrichment. (**B**) Susceptibility to infection: association between hGC and the odds of infection with *Babesia microti*, showing that females with the haplotype have an increased susceptibility to infection (from a GLMM). (**C**) Reproduction: association between hGC and reproductive success, showing that males with the haplotype have lower reproductive success (from a GLM) (error bars represent ± standard error; *p<0.05; **p<0.01; see *Table 1* for sample sizes). GLM = generalized linear model; GLMM = generalized linear mixed effects model.

that could be made to a pathogen in vivo (see 'Materials and methods' for details). Although our panel of genes did not include *Il33* or *Socs3*, it did include other genes associated with the inflammatory response, including *Il17a*, *Ifng*, *Il1b*, *Il6*, and *Tnfa*. We found that the effect of the GC haplotype on the stimulated expression levels of *Il17a*, a pro-inflammatory cytokine involved in the antibacterial response, displayed a significant interaction with sex (genotype by sex interaction in follow-up linear mixed effects model [LMM], likelihood ratio test [LRT] p=0.001). While females with the GC haplotype expressed lower levels of *Il17a* (estimate = −0.12, 95% CI = −0.19 to −0.04), males with the GC haplotype did not differ in their expressed levels of *Il17a* (estimate = 0.03, 95% CI includes zero = −0.02 to 0.08; *Appendix 1—table 6* and *Appendix 1—table 7*).

## The GC haplotype affects oxidative stress

Inflammation and oxidative stress are closely related and often lead to pathology (*Biswas, 2016*). To test whether the effects of the GC haplotype on inflammation may be having knock-on effects on oxidative stress, we again used the cross-sectional component of our study. We tested the effect of the GC haplotype on antioxidant superoxide dismutase 1 (SOD1) enzymatic activity in males and females. SOD1 is one of the main antioxidative enzymes within the antioxidative enzyme system (*Sun et al., 1989*). We found that (irrespective of sex) individuals with the AT haplotype had lower SOD1 activity compared with other haplotypes, including the GC haplotype (p=0.04; *Appendix 1—table 8*).

## The GC haplotype affects susceptibility to *Babesia* infection in females

We then used the longitudinal component of our study to test the effects of the GC haplotype on the probability of infection with common parasites known to infect our study population: two microparasites *Babesia microti* and *Bartonella* spp. and a range of macroparasites (ticks, fleas, and cestodes). We found an effect of the GC haplotype on the probability of infection with *B. microti*, which displayed

a marginally significant interaction with sex (genotype by sex interaction, LRT p=0.048; *Appendix 1—table 9*). While females with the GC haplotype were more likely to be infected with *B. microti* (log odds ratio = 1.25, 95% CI = 0.44–2.07; *Figure 1B*, lower panel), males with the GC haplotype did not differ in their susceptibility to infection (log odds ratio = 0.14, 95% CI includes zero = –0.76 to 1.03; *Figure 1B*, upper panel). However, we found no effect of the haplotype (interactive or not) on the probability of infection with the other parasites in our population (*Appendix 1—table 10*, *Appendix 1—table 11* and *Appendix 1—table 17*).

## The GC haplotype reduces male reproductive success

We genotyped both cross-sectional and longitudinal voles for 114 SNPs and used this dataset to construct a pedigree. We estimated each individual's reproductive success by counting the number of offspring it produced according to this pedigree. We then tested the effects of the GC haplotype on this measure of reproductive success. We tried to run a single model with both sexes (as above) but resulting residual variances differed significantly between the sexes, which reflects the fact that male and female reproduction are inherently different traits. This made it impossible to formally test for a genotype by sex interaction, and so instead we ran a separate model for each sex. We found that males with the GC haplotype had an average of 2.2 fewer offspring than males without the haplotype (p=0.04; *Appendix 1—table 12*; *Figure 1C*, upper panel). Despite a larger sample size and lower variability in reproductive success, females with the GC haplotype did not differ significantly in their number of offspring (genotype term did not appear in best model; *Appendix 1—table 13*; *Figure 1C*, lower panel). This is suggestive of a significant genotype by sex interaction. We ran the same models including microparasite variables, but this had little effect on our results (see *Appendix 1—table 14* and *Appendix 1—table 15*), suggesting that these effects were not driven by differences in microparasite infection.

## Discussion

In this study, we describe a polymorphism in the high-affinity receptor for IgE with effects that act across immune gene expression, oxidative stress, susceptibility to infection, and ultimately reproductive success in a natural population. This begins to reveal how genotypic effects can have multiple effects on different phenotypic or life history traits for organisms living in the natural environment. Interestingly, effects often differed between sexes, with evidence for opposing effects in the sexes (unstimulated immune gene expression) and an effect present in one sex but not the other (stimulated immune gene expression, susceptibility to infection, and reproductive success). However, we cannot rule out that, in the case of susceptibility to infection and reproductive success, there was a smaller effect present in the opposite sex (in the same or opposite direction), which our analysis did not have sufficient power to detect.

In humans, naturally occurring polymorphisms in the high-affinity IgE receptor have previously been linked to inflammatory disease (*Hasegawa et al., 2003*; *Potaczek et al., 2006*; *Zhou et al., 2012*; *Niwa et al., 2010*). Our work is consistent with this and suggests differing effects of this polymorphism on the inflammatory phenotype of males and females. While males with the GC haplotype showed a pro-inflammatory bias, females with the haplotype showed an anti-inflammatory bias. A previous study on another polymorphism in the human ortholog of *Fcer1a* also found evidence for sex-dependent effects on inflammatory disease (*Yang et al., 2013*). In order to fully understand this genotype by sex interaction, it is important to consider background differences in the inflammatory phenotypes of males and females. Females in this study were pregnant for a large proportion of their lives (57% of post-juvenile females were pregnant or lactating at the time of capture in the longitudinal component of our study). In mammals, including humans, pregnancy has been shown to be a largely anti-inflammatory state, with pregnant females suppressing inflammation. This is to protect the fetus from attack by the mother's immune system (*Wegmann et al., 1993*). While testosterone in males has an anti-inflammatory effect (*Bianchi, 2019*), it also drives males to use more space (*Davis et al., 2015*) and to be more aggressive (*Martínez-Sanchis et al., 2003*). These behaviors might increase their rates of contact with other individuals, and with their environment, and hence their likelihood of infection and subsequent immune stimulation. We suggest that the polymorphism may be exaggerating background sex differences in inflammatory activity. Sex hormones have been implicated in the differential

expression of some autosomal genes in males and females (*Mayne et al., 2016*), and could be driving these opposing effects. Some of the differences in immune phenotype that we observed may also be driven by difference in parasite infection (although we accounted for cestode burden in a follow-up analysis, we cannot rule this out).

The effect of *Fcer1a* polymorphism on inflammatory responses may also, in turn, affect individual susceptibility to infection. This is consistent with a previous association found between IgE-mediated responses and defense against helminths (*Gounni et al., 1994*). Although we did not find an association between macroparasite burden (including cestode burden) and the GC haplotype in this study, in our previous study, we showed that males with the GC haplotype had a lower level of an immunological marker of tolerance to ticks, fleas, and adult cestodes (i.e., they were less tolerant) (*Wanelik et al., 2018*). Here, we use data on infection incidence to show that, in a different context, the same haplotype is also associated with resistance to a microparasite, *B. microti*, in females. Although observational studies of this kind are not able to identify causal relationships, a realistic scenario is that females with the GC haplotype are more likely to be infected with *B. microti* (i.e., they are less resistant) as a result of a lower pro-inflammatory to anti-inflammatory cytokine ratio. Pro-inflammatory cytokines (e.g., IL-6, IFN-γ, and TNF-α) may help to resist *B. microti* infection (*Djokic et al., 2018*). A lack or imbalance of these cytokines may hamper this resistance. The panel of parasites that we have measured is not exhaustive, and previous studies have highlighted the important role of species interactions in the parasite community in driving infection risk in this study population (*Telfer et al., 2010*). *B. microti* may therefore represent a community of parasites, to which this haplotype affects resistance in females.

A fitness consequence of the GC haplotype was observed in reduced male reproductive success. A realistic scenario is that the larger cost incurred by males with the GC haplotype due to inflammation reduced their reproductive value, as reproducing and mounting a pro-inflammatory response are both costly activities (*Sheldon and Verhulst, 1996*). Another realistic scenario is that the positive effect of the GC haplotype on levels of oxidative stress (as indicated by a tendency for higher SOD1 activity) may have had a more detrimental effect in males, damaging sperm and reducing fertilizing success. Sperm competition can be an important feature in the reproduction of microtines, for example, meadow vole (*M. pennsylvanicus*) (*Delbarco-Trillo and Ferkin, 2004*), with males having to produce many sperm of high quality in order to successfully outcompete other males. Sperm are also highly susceptible to damage by reactive oxygen species (ROS) (*Losdat et al., 2011*). In addition, the heightened level of oxidative stress may have impaired olfactory sexual signals, for example, excretion of major urinary proteins (MUPs) making males less attractive to females and negatively impacting on mating success (*Garratt et al., 2014*).

Despite the immune system playing an important role in determining the health and reproductive fitness of an individual, natural selection has not converged on a single immune optimum. Instead, individuals in natural populations vary widely in their response to infection. Understanding why immunoheterogeneity is maintained is a key question in eco-immunology. Antagonistic selection, where a mutation is beneficial in some environmental contexts and harmful in others, is thought to be one mechanism by which balancing selection is generated, and genetic variation in immunity can be maintained within natural populations (*Graham et al., 2010*). In samples collected between 2008 and 2010, we previously identified four haplotypes at this locus, GC, AC, AT, and GT, at frequencies of 0.12, 0.76, 0.07, and 0.05, respectively (*Wanelik et al., 2018*). Seven years on, these frequencies have remained relatively unchanged (0.08, 0.81, 0.10, and 0.01). The fact that the GC haplotype remains in the population, despite its detrimental effects on male reproductive success and female infection susceptibility, suggests that it may be under balancing selection.

In order to be under balancing selection though, the GC haplotype would need to confer some advantage. We were not able to find evidence for any such advantage in this study, but we can speculate on one. Microtines show cyclical population dynamics, alternating between a 'peak' phase (associated with good conditions and high vole densities) and a 'crash' phase (poor conditions and low vole densities). Selection pressures acting during these two phases will be very different. We found that the GC haplotype was associated with a pro-inflammatory bias in males, indicating higher investment in promoting inflammation. Thus, one possible scenario is that higher investment in promoting inflammation is disadvantageous in the 'peak' phase when the risk of parasitism is low due to dilution effects, and it pays not to waste energy mounting a stronger immune response that is unnecessary.

However, it might be advantageous in the 'crash' phase when conditions are poor, and the risk of parasitism is high due to the time-lagged transmission of parasite infectious stages that have accumulated during the peak phase. Vole numbers were extremely low in our study area between 2012 and 2013 (indicating a 'crash phase'). In 2014, vole numbers began to increase and by the beginning of 2015 (the start of our study) had reached a peak. Although vole numbers declined throughout our study period, they never reached the extremely low numbers recorded in 2012–2013 (X. Lambin, personal communication, 2016). This suggests that we sampled in the region of a 'peak' phase, when this investment was disadvantageous, and that if we sampled during a true 'crash' phase we might detect an advantage to the GC haplotype in males.

There is a growing interest in human genomic (or precision) medicine, with the potential to use a patient's genotypic information to personalize their treatment. What we have shown here demonstrates that considering genotype in isolation can be misleading as the same polymorphism can have different outcomes not only for the immune gene expression, but the susceptibility to infection and ultimate reproductive success of males and females.

## Materials and methods

We studied *M. agrestis* in Kielder Forest, Northumberland, UK, using live-trapping of individual animals from natural populations. Trapping was performed from 2015 to 2017 across seven different sites, each a forest clear-cut. Access to the sites was provided by the Forestry Commission. At each site, 150–197 Ugglan small mammal traps (Grahnab, Gnosjo, Sweden) were laid out in a grid, spaced approximately 3–5 m apart. Our study was divided into longitudinal and cross-sectional components.

### Longitudinal data

Every other week, traps were checked every day, once in the morning and once in the evening. Newly trapped field voles were injected with a Passive Integrated Transponder (PIT) tag (AVID, Uckfield, UK) for unique identification. This approach allowed us to build up a longitudinal record for voles that were caught on multiple occasions. A total of 850 voles were individually marked in this way. We also took a small tissue sample from the tail for genotyping of individuals and a drop of blood from the tail which we put into 500 µl of RNAlater (Fisher Scientific, Loughborough, UK) for use in parasite detection (see below).

#### Parasite detection

We quantified infections by microparasites (*B. microti* and *Bartonella* spp.) in blood samples taken from longitudinal animals using SYBR green based two-step reverse transcription quantitative PCR (Q-PCR) targeted at pathogen ribosomal RNA genes. Expression values were normalized to two host endogenous control genes: *Ywhaz* and *Actb*. Blood samples were derived from tail bleeds. RNA was extracted from blood samples stored in RNAlater at –70°C using the Mouse RiboPure Blood RNA Isolation Kit (Thermo Fisher, Waltham, MA) according to the manufacturer's instructions and DNAse treated. It was then converted to cDNA using the High-Capacity RNA-to-cDNA Kit (Thermo Fisher), according to the manufacturer's instructions. *B. microti* primer sequences targeting the 18S ribosomal RNA gene were as follows CTACGTCCCTGCCCTTTGTA (forward primer sequence) and CCACGTTTCTTGGTCCGAAT (reverse primer sequence). *Bartonella* spp. primer sequences targeting the 16S ribosomal RNA gene were as follows: GATGAATGTTAGCCGTCGGG (forward primer sequence) and TCCCCAGGCGGAATGTTTAA (reverse primer sequences). Assays were pipetted onto 384-well plates by a robot (Pipetmax; Gilson, Middleton, WI) using a custom program and run on a QuantStudio 6-flex Real-Time PCR System (Thermo Fisher) at the machine manufacturer's default real-time PCR cycling conditions. Reaction size was 10 µl, incorporating 1 µl of template (diluted 1/20) and PrecisionFAST qPCR Master Mix (PrimerDesign, Chandler's Ford, UK) with low ROX and SYBR green and primers at the machine manufacturer's recommended concentrations. Alongside the pathogen assays, we ran assays for two host genes (see above) that acted as endogenous controls. For the pathogen assays, we used as a calibrator sample a pool of DNA extracted from 154 blood samples from different *M. agrestis* at our study sites in 2015 and 2016; these DNA extractions were carried out using the QIAamp UCP DNA Micro Kit (QIAGEN, Hilden, Germany) following the manufacturer's instructions. For the *Ywhaz* and *Actb* assays, the calibrator sample was the same cDNA calibrator sample as

described below for other host gene expression assays. Relative expression values (normalized to the two host endogenous control genes and presumed to relate to the expression of pathogen ribosomal RNA genes and in turn to the force of infection) used in analyses below are RQ values calculated by the QuantStudio 6-flex machine software according to the ΔΔCt method, indexed to the calibrator samples. Melting curves and amplification plots were individually inspected for each well replicate to confirm specific amplification. We validated our diagnostic results by comparing our PCR RQ values to independent data for a subset of voles from the cross-sectional component of our study with mapped genus-level pathogen reads from RNASeq analysis of blood samples (n = 44), finding that the two datasets strongly corroborated each other (*Bartonella* spp., Spearman's $\rho$ = 0.79, p<0.001, n = 44; *B. microti*, Spearman's $\rho$ = 0.88, p<0.001, n = 43).

## Cross-sectional data

For the cross-sectional component of the study (which ran from 2015 to 2016 only), animals were captured and returned to the laboratory where they were killed by a rising concentration of $CO_2$, followed by exsanguination. As our UK Home Office license did not allow us to sample overtly pregnant females in this way, there are fewer females than males present in this dataset. This component of the study allowed us to take a more comprehensive set of measurements and to culture cells and perform stimulatory assays on them. A small tissue sample was taken from the ear of cross-sectional animals for genotyping (see below).

### Parasite detection

After culling, the fur of cross-sectional animals was examined thoroughly under a binocular dissecting microscope to check for ectoparasites, which were recorded as direct counts of ticks and fleas. Guts of cross-sectional animals were transferred to 70% ethanol, dissected, and examined under a microscope for gastrointestinal parasites. Direct counts of cestodes (by far the dominant endohelminths in biomass) were recorded.

### SOD1 measurement

Given the well-established link between inflammation, oxidative stress, and pathology (*Biswas, 2016*), we measured antioxidant enzymatic activity in blood samples taken from cross-sectional animals. We chose to measure superoxide dismutase 1 (SOD1) because it is one of the main antioxidative enzymes within the antioxidative enzyme system (*Sun et al., 1989*) and is clearly linked to changes in immune gene expression in laboratory mice (*Marikovsky et al., 2003*). Assays were carried out using the Cayman Superoxide Dismutase kit and following the manufacturer's instructions except where otherwise indicated below. Blood pellets from centrifuged cardiac bleeds were stored at –70°C and thawed on ice prior to assay. A 20 µl aliquot from each pellet was lysed in 80 µl of ultrapure water and centrifuged (10,000 × *g* at 4 °C for 15 min) and 40 µl of the supernatant added to a 1.6× volume of ice-cold chloroform/ethanol (37.5/62.5 [v/v]) (inactivating superoxide dismutase 2). This mixture was then centrifuged (2500 × *g* at 4°C for 10 min) and the supernatant removed and immediately diluted 1:200 in kit sample buffer. A seven-point SOD activity standard was prepared (0, 0.005, 0.010, 0.020, 0.030, 0.040, and 0.050 U/ml) and the assay conducted on a 96-well microplate with 10 µl of standard or sample, 200 µl of kit radical detector solution, and 20 µl of xanthine oxidase (diluted as per the manufacturer's instructions) per well. Plates were then covered with a protective film, incubated at room temperature for 30 min on an orbital shaker, and read at 450 nm on a VERSAmax tunable absorbance plate reader (Molecular Devices, San Jose, CA), subtracting background and fitting a linear relationship to the standard curve in order to estimate SOD activity in unknown samples.

### Splenocyte cultures

Spleens of cross-sectional animals were removed, disaggregated, and splenocytes cultured under cell culture conditions equivalent to those used in *Jackson et al., 2011*. Unstimulated splenocytes, taken from 84 cross-sectional animals collected between July and October 2015, were initially used to assay expression by RNASeq (see below). We exposed splenocytes from the remaining cross-sectional animals to stimulation with anti-CD3 antibodies (Hamster Anti-Mouse CD3e, Clone 500A2 from BD Pharmingen, San Diego, CA) and anti-CD28 antibodies (Hamster Anti-Mouse CD28, Clone 37.51 from Tombo Biosciences, Kobe, Japan) at concentrations of 2 µg/ml and of 1 µg/ml, respectively, for 24 hr.

Costimulation with anti-CD3 and anti-CD28 antibodies was used to selectively promote the proliferation of T-cells (*Frauwirth and Thompson, 2002*; *Wanelik et al., 2020*). We assumed that this would reflect the potential response of T-cell populations in vivo. Stimulated splenocytes were used to assay expression by Q-PCR.

## RNASeq

Full details of the methods used for RNA preparation and sequencing can be found in *Wanelik et al., 2018*. Briefly, samples were sequenced on an Illumina HiSeq4000 platform. High-quality reads were mapped against a draft genome for *M. agrestis* (GenBank accession no. LIQJ00000000) and counted using featureCounts (*Liao et al., 2014*). Genes with average log counts per million, across all samples, of less than one were considered unexpressed and removed from the data (n = 8410). Following filtering, library sizes were recalculated, data were normalized, and MDS plots were generated to check for any unusual patterns in the data. The mean library size was 19 million paired-end reads (range = 3–71 million paired-end reads).

## Q-PCR

We used SYBR green-based Q-PCR to measure the expression levels of a panel of 18 genes (*Appendix 1—table 16*) in splenocytes, from our cross-sectional animals, that had been stimulated with T-cell agonist anti-CD3 and anti-CD28 antibodies. We did this, in part, to validate our RNASeq results in an independent dataset. We used the observed expression profile as a general measure of the potential responsiveness of the immune system to an inflammatory stimulation in vivo. We note that the genetic focus of our study, *Fcer1a*, codes for the alpha chain protein of the high-affinity IgE receptor that is expressed on cells such as eosinophils, basophils, and mast cells (*Daeron and Nimmerjahn, 2014*). Although the high-affinity IgE receptor may not be expressed significantly by the T-cells preferentially stimulated in our cultures (but see *Schadt, 2009*), the principle that genetic variants functionally expressed in one cell type may affect the function of other cell types through molecular interaction networks is embodied in the modern understanding of network biology (*Schadt, 2009*; *Barabási et al., 2011*). Thus, the effect of polymorphism in *Fcer1a*, expressed in cells such as eosinophils, basophils, or mast cells, either in vivo prior to isolation, or in the ex vivo cultured splenocyte population, could act indirectly on T-cells through various pathways, including via cytokine signaling (*Rothenberg and Hogan, 2006*; *Akuthota et al., 2008*; *Villanueva, 2015*). For example, eosinophils are known to promote helper T-cell activation and proliferation in ex vivo co-culture (*Liu et al., 2006*; *Harfi et al., 2013*) and in in vivo models (*Shi et al., 2004*). The choice of our panel of genes was informed by (i) known immune-associated functions in mice, combined with (ii) significant sensitivity to environmental or intrinsic host variables in our previous studies (*Jackson et al., 2014*; *Jackson et al., 2011*) or in a recent DGE analysis of RNASeq data (not reported here), and (iii) the aim of limited redundancy, with each gene representing a different immune pathway.

Primers (20 sets, including 2 endogenous control genes) were designed de novo and supplied by PrimerDesign (16 sets) or designed de novo in-house (4 sets) and validated (to confirm specific amplification and 100 ± 10% PCR efficiency under assay conditions). All PrimerDesign primer sets were validated under our assay conditions before use. The endogenous control genes (*Ywhaz* and *Sdha*) were selected as a stable pairing from our previous stability analysis of candidate control genes in *M. agrestis* splenocytes (*Jackson et al., 2011*). We extracted RNA from splenocytes conserved in RNAlater using the RNAqueous Micro Total RNA Isolation Kit (Thermo Fisher), following the manufacturer's instructions. RNA extracts were DNAse treated and converted to cDNA using the High-Capacity RNA-to-cDNA Kit (Thermo Fisher), according to the manufacturer's instructions, including reverse transcription negative (RT-) controls for a subsample. SYBR green-based assays were pipetted onto 384-well plates by a robot (Pipetmax, Gilson) using a custom program and run on a QuantStudio 6-flex Real-Time PCR System (Thermo Fisher) at the machine manufacturer's default real-time PCR cycling conditions. Reaction size was 10 µl, incorporating 1 µl of template and PrecisionFAST qPCR Master Mix with low ROX and SYBR green (PrimerDesign) and primers at the machine manufacturer's recommended concentrations. We used two standard plate layouts for assaying, each of which contained a fixed set of target gene expression assays and the two endogenous control gene assays (the same sets of animals being assayed on matched pairs of the standard plate layouts). Unknown samples were assayed in duplicate wells and calibrator samples in triplicate wells, and no template controls for each

gene were included on each plate. Template cDNA (see above) was diluted 1/20 prior to assay. The calibrator sample (identical on each plate) was created by pooling cDNA derived from across all splenocyte samples. Samples from different sampling groups were dispersed across plate pairs, avoiding confounding of plate with the sampling structure. Gene relative expression values used in analyses are RQ values calculated by the QuantStudio 6-flex machine software according to the ΔΔCt method, indexed to the calibrator sample. Melting curves and amplification plots were individually inspected for each well replicate to confirm specific amplification.

## Longitudinal and cross-sectional data

### Genotyping

We genotyped both cross-sectional and longitudinal animals for 346 SNPs in 127 genes. See *Wanelik et al., 2018* for details of the approach used to select these SNPs. Our list included two synonymous and tightly linked ($r^2 = 0.50$; D' = 0.70) SNPs in the gene *Fcer1a* (the alpha chain of the high-affinity receptor for IgE) on scaffold 582 (CADCXT010006977 in ENA accession GCA_902806775; see *Appendix 1—table 1* for genomic location information), which we had previously identified as a candidate tolerance gene in a natural population of *M. agrestis*. In this previous work, we had identified four haplotypes at this locus present in our population: GC, AC, AT, and GT, at frequencies of 0.12, 0.76, 0.07, and 0.05, respectively. We had also identified the GC haplotype as being of particular interest, given its significantly lower expression level of the transcription factor GATA3 (a biomarker of tolerance to macroparasites in our population) compared to the other haplotypes (*Wanelik et al., 2018*). We concluded that this haplotype tagged a causal mutation in the coding sequence, or in the up- or downstream regulatory regions. DNA was extracted from a tail sample (longitudinal component) or an ear sample (cross-sectional component) taken from the animal using DNeasy Blood and Tissue Kit (QIAGEN). Genotyping was then performed by LGC Biosearch Technologies (Hoddesdon, UK; http://www.biosearchtech.com) using the KASP SNP genotyping system. This included negative controls (water) and duplicate samples for validation purposes. The resulting SNP dataset was used for two purposes: (i) genotyping individuals within the locus of interest and (ii) pedigree reconstruction (see below).

### Pedigree reconstruction

We used a subset of our SNP dataset to reconstruct a pedigree for both cross-sectional and longitudinal animals using the R package *Sequoia* (*Huisman, 2017*). SNPs that violated the assumptions of Hardy–Weinberg equilibrium were removed from the dataset. For pairs of SNPs in high linkage disequilibrium (most commonly within the same gene), the SNP with the highest minor allele frequency (MAF) was chosen. A minimum MAF cutoff of 0.1 and call rate of >0.7 was then applied, and any samples for which >50% of SNPs were missing were removed. This resulted in a final dataset including 114 SNPs – a sufficient number for very good performance of parentage assignment (*Huisman, 2017*).

Life history information, namely, sex and month of birth, was inputted into *Sequoia* where possible. Juvenile voles weighing less than 12 g on first capture were assigned a birth date 2 weeks prior to capture. Juvenile voles weighing between 12 and 15 g on first capture were assigned a birth date 4 weeks prior to capture. Finally, adult voles breeding on first capture were assigned a birth date 6 weeks prior to capture (minimum age at first breeding) (*Begon et al., 2009*; *Burthe et al., 2010*). Adult voles not breeding on first capture could not be assigned a birth date as it was not known whether they had previously bred or not. Virtually all samples (99%) were assigned a sex, and approximately half (54%) were assigned a birth month. As we sampled individuals from across seven different clear-cut areas of the forest, each several kilometers apart, these were assumed to be independent, closed populations with negligible dispersal. Site-specific pedigrees were therefore generated.

We assessed the accuracy of our reconstructed pedigrees by checking whether predicted parent–offspring pairs met expectations given the biology of *M. agrestis*. As expected, the majority of predicted parent–offspring pairs (87%) were born in the same year. We also expected parents and offspring to overlap in space. Again, as expected, the majority of predicted parent–offspring pairs (92%) were, at some point, found along the same transects (horizontal or vertical). We also inspected log10 likelihood ratios (LLRs) for parent pairs as recommended in the user manual for *Sequoia*. Almost all LLRs were positive (n = 698/720 or 97% of LLRs) indicating confidence in our assignments. Individuals with vs. without our haplotype of interest did not differ in their probability of appearing in a

**Table 1.** Model specifications including, for each main model, covariates included in the full model, datasets used, and sample sizes (F = included as a fixed effect; R = included as a random effect).

| | DGE analysis | Haplotype association analyses Response variable Immune gene expression | Parasite infection | Reproductive success | SOD1 activity |
|---|---|---|---|---|---|
| **Covariates** Snout-vent length | | F | | | F |
| Eye lens weight | | F | | | F |
| Reproductive status | | F | F | | F |
| Body condition | | F | F | | F |
| Birth month | | | | F | |
| Culled or not | | | | F | |
| Site | | F; R (*Il17a* LMM) | F (Macro); R (Micro) | F | F |
| Year | | F | F | F | F |
| Season | | F; R (*Il17a* LMM) | F (Macro); R (Micro) | | F |
| Individual ID | | | R (Micro) | | |
| Assay plate | | R (*Il17a* LMM) | | | |
| **Dataset** | C | C | C (Macro) L (Micro) | C+L | C |
| **Sample size** | ♀31 ♂53 | *Il17a* LMM: ♀73 ♂220 | Macro: ♀82 ♂235 *B. microti*: ♀1075 ♂1247 *Bartonella* spp.: ♀1283 ♂1104 | ♀419 ♂232 | ♀81 ♂227 |

C = cross-sectional; L = longitudinal; DGE = differential gene expression; LMM = linear mixed effects model.

pedigree ($\chi^2$ = 0.09, d.f. = 1, p=0.76). For each individual that ended up in a pedigree, that is, with one or more relatives recorded (n = 652; site COL = 3; site BLB = 125; site GRD = 204; site CHE = 137; site RAV = 16; site SCP = 90; site HAM = 77), the number of offspring was counted to provide a measure of their reproductive success. Few individuals were first trapped as juveniles, with the majority trapped as adults that had already recruited into the population. Our measure of reproductive success then more closely resembles the number of recruited (rather than newborn) offspring per individual. Half of the individuals present in our pedigrees (n = 325) were found to have no offspring. We expect the majority of these to be true zeros (representing actual reproductive failure) as we generally sampled a large proportion of the total population within clear-cuts. We also minimized the chance of false zeros by excluding those individuals (e.g., at the periphery of a study grid) that did not end up in a pedigree because we identified no relatives (including offspring), likely because we had not sampled in the right place.

## Statistical analyses

Not all individuals appeared in all datasets; therefore, sample sizes (reported in *Table 1*) vary between analyses. All analyses were performed in R statistical software version 3.5.2 (*R Development Core Team, 2018*).

### Differential gene expression analysis

DGE analysis was performed on filtered and normalized count data using the R package edgeR (*Robinson et al., 2010*), the aim being to identify individual genes that were differentially expressed between those individuals with and without a copy of the GC haplotype (i.e., a dominant model). Only those individuals for which haplotype could be inferred with certainty could be included (n = 53 males

and n = 31 females; none of which were known to have two copies of the GC haplotype, hence the choice of a dominant model). Samples from different sexes were analyzed together. To test whether the top differentially expressed genes differed between the sexes, we included a dummy variable with four levels (males with haplotype, males without haplotype, females with haplotype, females without haplotype) and inspected the contrasts of interest (males with versus without haplotype; females with versus without haplotype). As this was an exploratory analysis, used in conjunction with more targeted measurements of immune gene expression (see below), model specification was kept as simple as possible and no covariates were initially included. However, in a follow-up DGE analysis we controlled for a key parasite variable (cestode burden).

Having confirmed a sex-dependent effect of the GC haplotype on the expression of some inflammatory genes (see 'Results') in the initial DGE analysis, we ran separate DGE analyses for males and females and tested more broadly for enrichment of pro- and anti-inflammatory genes in our results; more specifically, the Gene Ontology terms 'positive regulation of inflammatory response' (GO:0050729; n = 143) and 'negative regulation of inflammatory response' (GO:0050728; n = 149). The aim here was to answer the question: are pro- or anti-inflammatory genes more highly ranked relative to other genes, when we compare individuals with and without the GC haplotype, and does this also differ between the sexes? This GSEA was performed using the R package limma (*Ritchie et al., 2015*), and genes were ranked on log fold change.

## Haplotype association analyses

Following the exploratory DGE analysis, the GC haplotype was tested for associations with (i) gene expression assayed by Q-PCR to validate these results, (ii) macro- and microparasite infection to test whether the GC haplotype predicted an individual's susceptibility to infection, and (iii) reproductive success to test whether the GC haplotype was associated with any reproductive costs. Given the well-established link between inflammation and oxidative stress (*Reuter et al., 2011*; *Collins, 1999*), we also tested for an association between the GC haplotype and SOD1 activity.

For all analyses, this was initially attempted using the R package hapassoc (*Burkett et al., 2006*) in order to maximize sample size (see below). Hapassoc infers haplotypes on the basis of data from single SNPs and allows likelihood inference of trait associations with resulting SNP haplotypes and other attributes. It adopts a generalized linear model framework and estimates parameters using an expectation–maximization algorithm. Hapassoc models assumed an additive genetic model. If the haplotype combination of an individual cannot, with certainty, be inferred from its genotyping data (i) because it is heterozygous at two or more markers or (ii) because it has missing data for a single marker, the approach implemented in hapassoc is to consider all possible haplotype combinations for that individual. Standard errors accounting for this added uncertainty are calculated using the *Louis, 1982* method. We compared the GC haplotype against the other two major haplotypes (AC and AT). Another haplotype, the GT haplotype, was identified in the population but this was present at such low frequencies (frequency ≤ 0.01 among individuals for which haplotype could be inferred with certainty) that it was omitted from all analyses. Results reported in the text for macroparasites and SOD1 activity come from these hapassoc models.

However, there are some restrictions on model specification within hapassoc (e.g., random terms cannot be included, limited choice of error distributions), so this was followed up with regression models for some analyses. As in the DGE analysis, these regression models only included those individuals whose haplotype combination could be inferred with certainty. Results reported in the text for gene expression assayed by Q-PCR, microparasites, and reproductive success come from these regression models. Again, as in the DGE analysis, genotype was coded as the presence or absence of the GC haplotype (i.e., a dominant model). Regression models were run using the R package lme4 (*Bates et al., 2015*) or glmmADMB (*Skaug et al., 2016*; *Fournier et al., 2012*).

*Table 1* provides a summary of (hapassoc or regression) model specifications. As we found evidence for sex-dependent effect in the initial DGE analysis, with the exception of the model for reproductive success (see below), all models included a genotype by sex interaction that was retained if it improved model fit. All models also accounted for other biological and technical covariates (the choice of which was informed by our previous work, the literature, and/or our experimental design). Full regression models (including all covariates and interactions of interest) were reduced using backward stepwise deletion of nonsignificant terms to minimize Akaike's information criterion (AIC), following the drop1

function. This was not possible for hapassoc models. Summary tables, including all estimates, standard errors, and *z*-statistics, are included in Appendix 1. Associated significance values are also included in these tables, except for LMMs and GLMMs, for which significance values were based on LRTs and are reported in the main text. Throughout, residuals from regression models were checked for approximate normality and homoscedasticity, and all covariates were tested for independence using variance inflation factors (all VIFs < 3).

## Association between GC haplotype and immune gene expression assayed by Q-PCR

As we ran a total of 18 hapassoc models (one model per gene), the Benjamini and Hochberg method of correction was applied to all p-values (*Benjamini and Hochberg, 1995*). Resulting *q*-values (FDR-corrected p-values) are reported, alongside original p-values (*Appendix 1—table 6*). A hapassoc model including a genotype by sex interaction was tested against a null model without an interaction term. A Gamma error distribution and log link were used. Other covariates considered potential drivers of immune gene expression and included in both models were informed by our previous work (*Jackson et al., 2011*). These included snout-vent length (SVL), eye lens weight (categorized into seven intervals; SVL and eye lens weight capture the combined influence of age and historical growth trajectory), reproductive status (males were considered to be reproductively active if they had descended testes; females if they were pregnant or had perforate vaginas), and body condition (estimated by regressing body weight against life history stage, SVL, and its quadratic term). Site, year, and season (four levels, designated as spring [March and April], early summer [May and June], late summer [July and August], and autumn [September and October]) were included to account for any spatial and/or temporal autocorrelation. All covariates were included as fixed effects. We did not use a multidimensional approach (such as principal component analysis) because of limited redundancy in our panel of genes.

In order to confirm these results, an LMM was run for a single immune gene for which expression appeared to be associated with genotype (*q* = 0.037). The LMM included random terms for site and season, as well as assay plate number (*Table 1*). The latter was included to account for nonindependence due to immunological assaying structure. All other covariates were the same as the hapassoc model. A Yeo-Johnson transformation (with $\lambda$ = –2) was used to achieve more normal and homoscedastic residuals (*Yeo and Johnson, 2000*).

## Association between GC haplotype and parasite infection

The three macroparasite measurements taken from cross-sectional animals (counts of ticks, fleas, and cestodes) were log-transformed ($\log_{10}$ [*x*+1]) and summarized as a single principal component (explaining 39% of total variation; *Appendix 1—table 17*) to avoid difficulties in interpretation due to multiple testing. See *Wanelik et al., 2018* and *Jackson et al., 2014* for full details of this approach. This combined measure of macroparasite burden was modeled using a hapassoc model with a Gaussian error distribution.

Microparasite infection status was assessed multiple times for the majority of individuals in the longitudinal component of the study (mean = 2.8; range = 1–11). Due to these repeated measures, microparasite infection could not be modeled using hapassoc. Instead, to test for an association between the GC haplotype and the probability of an individual being infected with a microparasite, we ran a GLMM with a binary response (infected or not), log link, and a random term for individual. Other covariates, considered potential drivers of both macro- and microparasite infection, were, again, informed by our previous work (*Wanelik et al., 2018*; *Jackson et al., 2014*; *Taylor et al., 2018*). These included body condition, reproductive status, year, season, and site (*Table 1*). Season and site were included as random terms in the GLMM for microparasites, as was individual identity, to account for nonindependence due to repeat sampling.

## Association between GC haplotype and reproductive success

Our measure of reproductive success was zero-inflated (50% zeros). This is consistent with a previous study of the closely related common vole (*Microtus arvalis*) (*Wang et al., 2019*). A Poisson error distribution was therefore deemed inappropriate, and it could not be modeled using hapassoc. Instead, to test for an association between the GC haplotype and reproductive success, we ran a

GLM with a quasi-Poisson error distribution. This distribution has been previously used to model predictors of reproductive success in other organisms, and accounts for the overdispersion caused by excess zeros.

We only recorded a reproductive failure (i.e., zero reproductive success) for those individuals with some other familial relationship in our pedigree, purposefully omitting those individuals (e.g., at the periphery of a study grid) for which we may have recorded no relatives (including offspring) simply because we had not sampled in the right place. Therefore, we expect all (or most) of our zeros to represent actual reproductive failure. For this reason, zero-inflated models were deemed inappropriate.

As detailed in the 'Results,' we tried to run a single quasi-Poisson GLM with both sexes but resulting residual variances differed significantly between the sexes (F test to compare variances of two samples; p=0.02), making it impossible to formally test for a genotype by sex interaction. Instead, we ran a separate model for each sex. We included birth month as a covariate in this model, given that autumn-born voles (of the closely related common vole) have been shown to have a lower chance to reproduce than spring-born voles (*Wang et al., 2019*). Other covariates included in this model were whether or not an individual was culled for the cross-sectional component of this study (again, reducing the opportunity to reproduce), site, and the year in which an individual was most frequently captured (*Table 1*). In a follow-up analysis, we also controlled for *B. microti* and *Bartonella* spp. infection status. More specifically, we included the proportion of samples taken from an individual that were *Babesia*-positive and the proportion of samples taken from an individual that were *Bartonella*-positive. All covariates were included as fixed effects. Only a single female was trapped in one of the sites (COL) and consequently caused convergence problems in the female model. This female was therefore omitted.

## Association between GC haplotype and SOD1 activity

SOD1 activity was modeled using a hapassoc model with a Gaussian error distribution. Other covariates, considered potential drivers of SOD1 activity, were informed by the literature. Previous studies on wild rodents have shown that antioxidant levels increase with both reproductive effort (*Garratt et al., 2011*) and with age (*Hindle et al., 2010*). Studies on birds have also shown that improvements in body condition are often accompanied by increases in antioxidant activity, for example, in response to supplemental feeding (*Wilcoxen et al., 2015*). We therefore included SVL, eye lens weight, reproductive status, and body condition as covariates in our model. As in other models, we included site, year, and season to account for spatial and/or temporal autocorrelation in our data (*Table 1*). All covariates were included as fixed effects.

# Acknowledgements

We thank all those involved in obtaining and processing samples from the field: Rebecca Turner, Lukasz Lukomski, Stephen Price, Sarah Gore, Ed Parker, Maria Capstick, Noelia Dominguez Alvarez, Susan Withenshaw, William Foster, Ann Lowe, and Benoit Poulin. We would also like to thank the Forestry Commission for access to the study sites and the Centre for Genomic Research at the University of Liverpool for sequencing samples.

# Additional information

### Funding

| Funder | Grant reference number | Author |
|---|---|---|
| Natural Environment Research Council | NE/L013452/1 | Mike Begon<br>Janette E Bradley<br>Joseph A Jackson<br>Steve Paterson |

The funders had no role in study design, data collection and interpretation, or the decision to submit the work for publication.

## Author contributions
Klara M Wanelik, Formal analysis, Writing – original draft, Writing – review and editing; Mike Begon, Janette E Bradley, Joseph A Jackson, Steve Paterson, Conceptualization, Funding acquisition, Writing – review and editing; Ida M Friberg, Investigation; Christopher H Taylor, Investigation, Writing – review and editing

## Author ORCIDs
Klara M Wanelik ⓘ http://orcid.org/0000-0003-1485-0340
Mike Begon ⓘ http://orcid.org/0000-0003-1715-5327
Janette E Bradley ⓘ http://orcid.org/0000-0003-3973-7977
Joseph A Jackson ⓘ http://orcid.org/0000-0003-0330-5478
Christopher H Taylor ⓘ http://orcid.org/0000-0003-4299-7104
Steve Paterson ⓘ http://orcid.org/0000-0002-1307-2981

## Ethics
All animal procedures were performed with approval from the University of Liverpool Animal Welfare Committee and under a UK Home Office licence (PPL 70/8210 to S.P.).

## Decision letter and Author response
Decision letter https://doi.org/10.7554/eLife.77666.sa1
Author response https://doi.org/10.7554/eLife.77666.sa2

## Additional files

### Supplementary files
• MDAR checklist

### Data availability
RNASeq data have been deposited in the European Nucleotide Archive (study accession number PRJEB51626). Longitudinal phenotypic data are available from the NERC EDS Environmental Information Data Centre (https://doi.org/10.5285/e5854431-6fa4-4ff0-aa02-3de68763c952). Cross-sectional phenotypic data are available from the University of Liverpool's Research Data Catalogue (https://doi.org/10.17638/datacat.liverpool.ac.uk/1850). Genotype data are also available from the University of Liverpool's Research Data Catalogue (https://doi.org/10.17638/datacat.liverpool.ac.uk/1849).

The following datasets were generated:

| Author(s) | Year | Dataset title | Dataset URL | Database and Identifier |
| --- | --- | --- | --- | --- |
| Paterson S | 2022 | Genotypes of field voles from Kielder Forest | https://datacat.liverpool.ac.uk/1849/ | University of Liverpool's Research Data Catalogue, 10.17638/datacat.liverpool.ac.uk/1849 |
| Paterson S | 2022 | Phenotypic and immunological data of field voles | https://datacat.liverpool.ac.uk/1850/ | University of Liverpool's Research Data Catalogue, 10.17638/datacat.liverpool.ac.uk/1850 |
| Paterson S | 2022 | RNAseq analysis of field voles (Microtus agrestis) | https://www.ebi.ac.uk/ena/browser/view/PRJEB51626 | European Nucleotide Archive, PRJEB51626 |
| Paterson S, Jackson J, Jackson I, Bradley J, Begon M, Wanelik K, Taylor C | 2022 | Data on immunological expression and phenotypes in a natural population of field voles in Kielder Forest, UK 2015-2017 | https://catalogue.ceh.ac.uk/documents/e5854431-6fa4-4ff0-aa02-3de68763c952 | NERC EDS Environmental Information Data Centre, 10.5285/e5854431-6fa4-4ff0-aa02-3de68763c952 |

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

# Appendix 1

**Appendix 1—table 1.** Position of single-nucleotide polymorphisms (SNPs) and other key features in the *Fcer1a* gene.

Features lie in scaffold CADCXT010006977 within assembly GCA_902806775.

| Feature | Start | End |
|---|---|---|
| *Fcer1a* gene | 10745528 | 10745528 |
| Exon 5 | 10745528 | 10745881 |
| Exon 4 | 10746818 | 10747076 |
| Exon 3 | 10748609 | 10748864 |
| Exon 2 | 10750044 | 10750065 |
| Exon 1 | 10750465 | 10750642 |
| 5' UTR | 10750520 | 10750642 |
| CDS | 10750465 | 10750520 |
| CDS | 10750044 | 10750065 |
| CDS | 10748609 | 10748864 |
| CDS | 10746818 | 10747076 |
| CDS | 10745729 | 10745881 |
| 3' UTR | 10745528 | 10745729 |
| SNP 2 | 10748718 | 10748719 |
| SNP 1 | 10746846 | 10746847 |

**Appendix 1—table 2.** Top 10 annotated genes that were differentially expressed between males with vs. without the GC haplotype, including associated log fold changes (logFC), p-values, and *q*-values (false discovery rate [FDR]-corrected p-values).

| Gene | Protein | LogFC | p-value | q-value |
|---|---|---|---|---|
| *Snai3* | Snail family zinc finger 3 | 1.698 | $1.220 \times 10^{-9}$ | $1.574 \times 10^{-5}$ |
| *Pla2g4c* | Phospholipase A2, group IVC (cytosolic, calcium-independent) | 2.904 | $6.386 \times 10^{-9}$ | $4.120 \times 10^{-5}$ |
| *Il33* | Interleukin 33 | 2.756 | $1.474 \times 10^{-7}$ | $4.755 \times 10^{-4}$ |
| *Mmp13* | Matrix metallopeptidase 13 | 2.571 | $4.427 \times 10^{-7}$ | $1.142 \times 10^{-3}$ |
| *Uba7* | Ubiquitin-like modifier activating enzyme 7 | 0.771 | $5.979 \times 10^{-7}$ | $1.286 \times 10^{-3}$ |
| *Robo4* | Roundabout guidance receptor 4 | 0.892 | $3.738 \times 10^{-6}$ | $4.385 \times 10^{-3}$ |
| *Ttn* | Titin | 1.665 | $8.995 \times 10^{-6}$ | $7.254 \times 10^{-3}$ |
| *Flnc* | Filamin C, gamma | 1.554 | $1.245 \times 10^{-5}$ | $9.447 \times 10^{-3}$ |
| *Crb2* | Crumbs family member 2 | 1.327 | $1.567 \times 10^{-5}$ | $1.063 \times 10^{-2}$ |
| *Muc16* | Mucin 16 | 1.937 | $1.971 \times 10^{-5}$ | $1.156 \times 10^{-2}$ |

**Appendix 1—table 3.** Annotated genes that were differentially expressed ($q \leq 0.1$) between females with vs. without the GC haplotype, including associated log fold changes (logFC), p-values, and *q*-values (false discovery rate [FDR]-corrected p-values).

| Gene | Protein | LogFC | p-value | q-value |
|---|---|---|---|---|
| *Tldc1* | TBC/LysM associated domain containing 1 | 1.489 | $2.502 \times 10^{-8}$ | $3.228 \times 10^{-4}$ |
| *Peg3* | Paternally expressed 3 | 1.440 | $5.369 \times 10^{-5}$ | 0.049 |
| *Socs3* | Suppressor of cytokine signaling 3 | 1.071 | $6.061 \times 10^{-5}$ | 0.052 |

**Appendix 1—table 4.** Top 10 annotated genes that were differentially expressed between males with vs. without the GC haplotype when controlling for cestode burden, including associated log fold changes (logFC), p-values, and q-values (false discovery rate [FDR]-corrected p-values).

| Gene | Protein | LogFC | p-value | q-value |
|---|---|---|---|---|
| Il33 | Interleukin 33 | 3.091 | $2.473 \times 10^{-10}$ | $3.191 \times 10^{-6}$ |
| Snai3 | Snail family zinc finger 3 | 1.672 | $5.259 \times 10^{-9}$ | $3.393 \times 10^{-5}$ |
| Pla2g4c | Phospholipase A2, group IVC (cytosolic, calcium-independent) | 2.883 | $1.562 \times 10^{-8}$ | $6.721 \times 10^{-5}$ |
| Uba7 | Ubiquitin-like modifier activating enzyme 7 | 0.809 | $2.477 \times 10^{-8}$ | $7.990 \times 10^{-5}$ |
| Mmp13 | Matrix metallopeptidase 13 | 2.606 | $4.090 \times 10^{-7}$ | $8.796 \times 10^{-4}$ |
| Robo4 | Roundabout guidance receptor 4 | 0.892 | $8.160 \times 10^{-6}$ | $9.572 \times 10^{-3}$ |
| Ttn | Titin | 1.599 | $2.532 \times 10^{-5}$ | $1.951 \times 10^{-2}$ |
| Flnc | Filamin C, gamma | 1.493 | $2.721 \times 10^{-5}$ | $1.951 \times 10^{-2}$ |
| Crb2 | Crumbs family member 2 | 1.261 | $3.512 \times 10^{-5}$ | $2.151 \times 10^{-2}$ |

**Appendix 1—table 5.** Annotated genes that were differentially expressed ($q \leq 0.1$) between females with vs. without the GC haplotype when controlling for cestode burden, including associated log fold changes (logFC), p-values, and q-values (false discovery rate [FDR]-corrected p-values).

| Gene | Protein | LogFC | p-value | q-value |
|---|---|---|---|---|
| Tldc1 | TBC/LysM associated domain containing 1 | 1.489 | $2.890 \times 10^{-8}$ | $3.730 \times 10^{-4}$ |
| Socs3 | Suppressor of cytokine signaling 3 | 1.083 | $4.133 \times 10^{-5}$ | 0.041 |
| Peg3 | Paternally expressed 3 | 1.411 | $6.375 \times 10^{-5}$ | 0.055 |
| Ppp1r3c | Protein phosphatase 1, regulatory (inhibitor) subunit 3C | 1.662 | $8.798 \times 10^{-5}$ | 0.071 |

**Appendix 1—table 6.** Significance values from hapassoc models for expression of 18 genes (assayed by Q-PCR) in splenocytes.

Splenocytes were stimulated with anti-CD3 and anti-CD28 antibodies in order to promote the proliferation of T-cells. q-values (false discovery rate [FDR]-corrected p-values) are reported alongside original p-values for the genotype by sex interaction.

| Gene | p-value | q-value |
|---|---|---|
| Cd4 | 0.124 | 0.822 |
| Cd8a | 0.744 | 0.866 |
| Foxp3 | 0.499 | 0.845 |
| Gata3 | 0.563 | 0.845 |
| Il10 | 0.866 | 0.866 |
| Mpo | 0.173 | 0.822 |
| Tbx21 | 0.650 | 0.866 |
| Tgfb1 | 0.271 | 0.822 |
| Ifng | 0.399 | 0.822 |
| Il17a | 0.002 | 0.037 |
| Il1b | 0.749 | 0.866 |
| Il6 | 0.282 | 0.822 |
| Ms4a1 | 0.281 | 0.822 |

*Appendix 1—table 6 Continued on next page*

*Appendix 1—table 6 Continued*

| Gene | p-value | q-value |
|------|---------|---------|
| *Orai1* | 0.353 | 0.822 |
| *Tnfa* | 0.858 | 0.866 |
| *Il2* | 0.411 | 0.822 |
| *Apobr* | 0.857 | 0.866 |
| *Arg1* | 0.552 | 0.845 |

**Appendix 1—table 7.** Estimates, standard errors, and *z*-statistics from best LMM for Yeo-Johnson-transformed *Il17a* expression levels.

| | Estimate | SE | z |
|------|----------|-----|-----|
| (Intercept) | 0.107 | 0.023 | 4.61 |
| Genotype | –0.115 | 0.037 | –3.10 |
| Sex male | –0.055 | 0.019 | –2.89 |
| Year 2016 | 0.084 | 0.018 | 4.60 |
| Genotype × sex male | 0.143 | 0.044 | 3.25 |

**Appendix 1—table 8.** Effect sizes, standard errors, *z*-statistics, and associated significance from Gaussian hapassoc model for SOD1 activity.

| | Estimate | SE | z | p-value |
|------|----------|-----|-----|---------|
| (Intercept) | 0.614 | 1.514 | 0.406 | 0.685 |
| hAT | –0.453 | 0.223 | –2.032 | 0.042 |
| Reproductive status active | –0.112 | 0.266 | –0.422 | 0.673 |
| Sex male | 0.031 | 0.250 | 0.124 | 0.901 |
| SVL | 0.016 | 0.017 | 0.945 | 0.345 |
| Lens weight 2 | 0.568 | 0.531 | 1.070 | 0.284 |
| Lens weight 3 | 0.440 | 0.549 | 0.801 | 0.423 |
| Lens weight 4 | 0.445 | 0.587 | 0.758 | 0.448 |
| Lens weight 5 | 0.415 | 0.653 | 0.636 | 0.525 |
| Lens weight 6 | 0.301 | 0.752 | 0.400 | 0.689 |
| Lens weight 7 | –0.307 | 1.095 | –0.280 | 0.779 |
| Body condition | 0.020 | 0.030 | 0.677 | 0.498 |
| Site CHE | 1.429 | 0.431 | 3.316 | 0.001 |
| Site COL | 2.675 | 1.099 | 2.434 | 0.015 |
| Site GRD | 1.476 | 0.437 | 3.379 | 0.001 |
| Site RAV | 0.904 | 0.918 | 0.984 | 0.325 |
| Site SCP | 1.107 | 0.419 | 2.643 | 0.008 |
| Year 2016 | 0.593 | 0.242 | 2.457 | 0.014 |
| Season early summer | –0.392 | 0.389 | –1.007 | 0.314 |
| Season late summer | –0.098 | 0.266 | –0.369 | 0.712 |
| Season spring | –0.969 | 0.383 | –2.529 | 0.011 |

**Appendix 1—table 9.** Effect sizes, standard errors, and *z*-statistics from best binomial GLMM for probability of infection with *Babesia microti*.

|  | Estimate | SE | z |
|---|---|---|---|
| (Intercept) | –2.214 | 0.377 | –5.87 |
| Genotype | 1.254 | 0.416 | 3.02 |
| Sex male | 0.957 | 0.229 | 4.19 |
| Year 2016 | 1.260 | 0.277 | 4.55 |
| Year 2017 | 1.047 | 0.302 | 3.46 |
| Reproductive status active | 0.754 | 0.144 | 5.24 |
| Body condition | 0.044 | 0.016 | 2.76 |
| Genotype × sex male | –1.116 | 0.615 | –1.82 |

**Appendix 1—table 10.** Effect sizes, standard errors, and *z*-statistics from best binomial GLMM for probability of infection with *Bartonella* spp.

|  | Estimate | SE | z |
|---|---|---|---|
| (Intercept) | 0.236 | 0.369 | 0.64 |
| Year 2016 | 0.740 | 0.135 | 5.49 |
| Year 2017 | 2.280 | 0.175 | 13.01 |

**Appendix 1—table 11.** Effect sizes, standard errors, *z*-statistics, and associated significance from Gaussian hapassoc model for macroparasite infection summarized by a single principal component.

|  | Estimate | SE | z | p-value |
|---|---|---|---|---|
| (Intercept) | 0.050 | 0.331 | 0.152 | 0.879 |
| hAC | –0.102 | 0.130 | –0.786 | 0.432 |
| hAT | –0.108 | 0.158 | –0.682 | 0.495 |
| Sex male | –0.271 | 0.126 | –2.146 | 0.032 |
| Season early summer | –0.103 | 0.182 | –0.567 | 0.571 |
| Season late summer | 0.012 | 0.132 | 0.090 | 0.929 |
| Season spring | 0.327 | 0.176 | 1.854 | 0.064 |
| Reproductive status active | –0.222 | 0.119 | –1.868 | 0.062 |
| Body condition | –0.019 | 0.014 | –1.328 | 0.184 |
| Year 2016 | –0.473 | 0.124 | –3.810 | 0.000 |
| Site CHE | 0.714 | 0.226 | 3.157 | 0.002 |
| Site COL | 0.762 | 0.575 | 1.327 | 0.185 |
| Site GRD | 0.707 | 0.229 | 3.090 | 0.002 |
| Site RAV | 1.100 | 0.481 | 2.288 | 0.022 |
| Site SCP | 0.747 | 0.220 | 3.389 | 0.001 |

**Appendix 1—table 12.** Effect sizes, standard errors, *z*-statistics, and associated significance from best quasi-Poisson GLM for reproductive success in males.

|  | Estimate | SE | z | p-value |
|---|---|---|---|---|
| (Intercept) | 1.743 | 0.243 | 7.171 | $1.05 \times 10^{-11}$ |

*Appendix 1—table 12 Continued on next page*

*Appendix 1—table 12 Continued*

|  | Estimate | SE | z | p-value |
|---|---|---|---|---|
| Genotype | −0.484 | 0.237 | −2.042 | 0.042 |
| Year 2016 | 0.592 | 0.185 | 3.193 | 0.002 |
| Year 2017 | −0.437 | 0.199 | −2.193 | 0.029 |
| Birth month | −0.358 | 0.048 | −7.449 | $1.97 \times 10^{-12}$ |
| Culled yes | 0.422 | 0.150 | 2.809 | 0.005 |

**Appendix 1—table 13.** Effect sizes, standard errors, z-statistics, and associated significance from best quasi-Poisson GLM for reproductive success in females.

|  | Estimate | SE | z | p-value |
|---|---|---|---|---|
| (Intercept) | 1.137 | 0.264 | 4.298 | $2.15 \times 10^{-5}$ |
| Year 2016 | 0.520 | 0.211 | 2.466 | 0.014 |
| Year 2017 | −0.118 | 0.215 | −0.549 | 0.584 |
| Birth month | −0.298 | 0.047 | −6.391 | $4.42 \times 10^{-10}$ |

**Appendix 1—table 14.** Effect sizes, standard errors, z-statistics, and associated significance from best quasi-Poisson GLM for reproductive success in males when controlling for *Babesia microti* and *Bartonella* spp. infection.

|  | Estimate | SE | z | p-value |
|---|---|---|---|---|
| (Intercept) | 1.467 | 0.418 | 3.513 | $7.64 \times 10^{-4}$ |
| Genotype | −1.081 | 0.425 | −2.544 | 0.013 |
| Year 2016 | 0.238 | 0.279 | 0.854 | 0.396 |
| Year 2017 | −0.912 | 0.386 | −2.338 | 0.022 |
| Birth month | −0.464 | 0.091 | −5.090 | $2.69 \times 10^{-6}$ |
| Proportion of samples *Babesia*-positive | 0.849 | 0.276 | 3.076 | 0.003 |
| Proportion of samples *Bartonella*-positive | 0.997 | 0.440 | 2.268 | 0.026 |

**Appendix 1—table 15.** Effect sizes, standard errors, z-statistics, and associated significance from best quasi-Poisson GLM for reproductive success in females when controlling for *Babesia microti* and *Bartonella* spp. infection.

|  | Estimate | SE | z | p-value |
|---|---|---|---|---|
| (Intercept) | 1.483 | 0.305 | 4.867 | $3.20 \times 10^{-6}$ |
| Birth month | −0.310 | 0.073 | −4.229 | $4.38 \times 10^{-5}$ |
| Proportion of samples *Babesia*-positive | 0.569 | 0.247 | 2.308 | 0.0226 |

**Appendix 1—table 16.** Panel of 18 genes for which expression levels in splenocytes stimulated with anti-CD3 and anti-CD28 antibodies were measured using two-step reverse transcription quantitative PCR (Q-PCR).

| Gene | Protein |
|---|---|
| *Cd4* | T-cell surface glycoprotein CD4 |
| *Cd8a* | T-cell surface glycoprotein CD8 alpha chain |
| *Foxp3* | Forkhead box protein P3 |
| *Gata3* | GATA binding protein 3 |

*Appendix 1—table 16 Continued on next page*

*Appendix 1—table 16 Continued*

| Gene | Protein |
|------|---------|
| *Il10* | Interleukin-10 |
| *Mpo* | Myeloperoxidase |
| *Tbx21* | T-box transcription factor TBX21 |
| *Tgfb1* | Transforming growth factor beta 1 |
| *Ifng* | Interferon gamma |
| *Il17a* | Interleukin-17a |
| *Il1b* | Interleukin-1 beta |
| *Il6* | Interleukin-6 |
| *Ms4a1* | B-lymphocyte antigen CD20 |
| *Orai1* | Calcium release-activated calcium channel protein 1 |
| *Tnfa* | Tumor necrosis factor alpha |
| *Il2* | Interleukin-2 |
| *Apobr* | Apolipoprotein B receptor |
| *Arg1* | Arginase-1 |

**Appendix 1—table 17.** Loadings from principal component analysis summarizing infection by macroparasites (ticks, fleas, and cestodes).

| Macroparasite | PC1 |
|---------------|-----|
| Cestodes | –0.481 |
| Ticks | –0.592 |
| Fleas | –0.646 |

