## [Editor Report]

This study provides an unusually comprehensive analysis of the associations between polymorphism in an immune gene, the immunoglobulin E receptor Fcer1a, and immune responses, resistance to infection, and reproductive fitness in a wild rodent population. The investigators find that these effects appear to be sex-specific. This study provides critical observations of the possible consequences of immune polymorphisms in wild populations and should be of interest to immunologists, evolutionary biologists, and ecologists investigating genotype-phenotype relationships and potential life-history tradeoffs.

---

## [Decision Letter]

**Decision letter after peer review:**

Thank you for submitting your article "Sex-specific effects of an IgE polymorphism on immunity, susceptibility to infection and reproduction in a wild rodent" for consideration by *eLife*. Your article has been reviewed by 3 peer reviewers, and the evaluation has been overseen by a Reviewing Editor and Jos van der Meer as the Senior Editor. The reviewers have opted to remain anonymous.

Essential revisions:

1. Temper claims to match statistics.

2. Exclude in vitro T cell stimulation tests of the FCER1A SNP or provide better justification for the assay (please see reviewer comments below).

3. Investigate potential confounding of IL-33 expression involving GC haplotype, IgE, and parasite burdens in a revised statistical analysis. For example, one reviewer suggests IL-33 should be included as an explanatory alongside all potential confounding variables such as parasite burdens, and the resulting fitted model estimate / z score for IL-33 be used to evaluate its importance.

*Reviewer #2 (Recommendations for the authors):*

In general, I would recommend you tone down the claims about sex-specific effects; actually, the results presented here are of interest regardless of whether they are sex-specific or not. Given that such effects are at best marginally significant, I think this is something that is better addressed in the Discussion, rather than claimed in the title.

*Reviewer #3 (Recommendations for the authors):*

– Please revise and expand reporting of the immune gene expression data (Il-33 is a type 2 cytokine, not normally considered pro-inflammatory). Much of the discussion of inflammatory vs anti-inflammatory requires more nuance – there are, at a minimum, antibacterial (IL-17), antihelminthic (IL-33, GATA3), and anti-inflammatory (SOC3) pathways being flagged up, and it would also be important to test antiviral responses (interferon-stimulated genes).

– Covariates detailed in Table 1 lack infection variables. These would be essential adjustments because the hypothesis is that all else being equal (i.e., at identical infection burdens, etc.), GC carriers should be more pro-inflammatory (males) or anti-inflammatory (females).

– The differential expression of Il33 (and any other genes/pathways of interest) should be followed up with statistical analysis to determine whether it is driven by worm infection, reproductive status, or other potential confounders in addition to / rather than haplotype. A GLMM containing covariates detailed in Table 1 + infection data would seem appropriate.

– Likewise, the effect of GC on male (and female?) reproductive success should be adjusted for the same potential confounders, like infection level.

– Figure 1A is quite uninformative. Please consider representing genes and top enriched pathways in a manner that allows readers to interpret the data.

[Editors’ note: further revisions were suggested prior to acceptance, as described below.]

Thank you for resubmitting your work entitled "Effects of an IgE receptor polymorphism acting on immunity, susceptibility to infection and reproduction in a wild rodent" for further consideration by *eLife*. Your revised article has been evaluated by Jos van der Meer (Senior Editor) and a Reviewing Editor.

The manuscript has been improved but there are some remaining issues that need to be addressed.

(1) Reviewer 2 has pointed out that it is difficult to understand how Fcer1a could be linked to IL17a production. This finding should be backed in the manuscript by some reference to previous evidence, or it should be made clear that this result is unexpected and difficult to explain. Reviewer 2 also suggests removing the result, which is another option.

(2) Please see other suggestions by Reviewer 2 for improving clarity.

*Reviewer #2 (Recommendations for the authors):*

The authors have dealt with most of my comments in a satisfactory way, but there are still a few significant problems. In particular, the rationale for the experiment where T cells were stimulated ex vivo is highly unclear (see below for details).

The conclusions are now better matched by data, although there are still quite strong claims about sex-specific effects even though there are few formal tests of this (i.e. tests for sex-by-genotype effects). More problematic is that the justification (L 578-581) provided for the analysis of ex vivo stimulated T cells is very vague. The problem with this assay is as follows: I believe that in splenocytes, Fcer1a is mainly (only?) expressed on basophils and eosinophils; thus, an effect of a Fcer1a polymorphism needs to involve these cells in some way. In your assay you stimulated T cells (which are abundant in splenocytes) in a non-specific way. Of the 18 cytokines measured, you find an effect of Fcer1a polymorphism on expression of IL17a, which is produced by T cells (but not by eosinophils/basophils). The problem is that I just cannot see how the Fcer1a polymorphism (which should have an effect in basophils/eosinophils, but not in T cells) could be linked to IL17a production by T cells; this would need to involve signalling from activated T cells to basophils/eosinophils, and then Fcer1a polymorphism-dependent signalling from basophils/eosinophils back to T-cells to affect IL17a production; I guess it is not impossible, but I am unaware of anything like that. I think you need to provide a sound and specific rationale for this assay; just saying "it could be acting indirectly on T-cells through various pathways, including via cytokine signalling, following Fcer1a expression by other cells" (L578-581) is not enough.

The presentation of the DGE analysis and GSEA of unstimulated splenocytes is improved, but still a bit confusing (L 199-238): (i) I think it would be good to make clear in the Results that the DGE analysis and GSEA are based on comparing individuals with the GC haplotype vs all others (so that the reader does not have to go the MandM to find this), (ii) using the term "top-responding" (L 206) implies that you are measuring the transcriptional response to some treatment, while you are just comparing baseline expression of GC vs others in males and females (or have I misunderstood something?); use "differentially expressed" or something similar instead, (iii) similarly, using the term "upregulated" (L 231) implies you measured the transcriptional response to some treatment; use "more highly expressed in individuals with GC…" or similar.

When it comes to the ex vivo stimulation of T cells, I strongly suggest you remove these results (L240-254); they just don't make sense from an immunological point of view. Unfortunately, this is one of few analyses that actually shows a sex-by-genotype effects, so removing this would weaken the support for sex-specific effects of the Fcer1a polymorphism even further.

---

## [Author Response]

Essential revisions:1. Temper claims to match statistics.

We have removed ‘sex-specific’ from the title and substantially reworked the introduction and discussion to reflect the comments by Reviewer 2 that the results are of interest regardless of whether they are sex-specific. A more detailed response is given below.

2. Exclude in vitro T cell stimulation tests of the FCER1A SNP or provide better justification for the assay (please see reviewer comments below).

We have provided a justification for this assay – although *Fcer1a* is not expressed by T-cells themselves, polymorphism in this gene could be acting indirectly on T-cells through various pathways, including via cytokine signalling, following expression of *Fcer1a* by other cells. Again, please see more detailed response given below.

3. Investigate potential confounding of IL-33 expression involving GC haplotype, IgE, and parasite burdens in a revised statistical analysis. For example, one reviewer suggests IL-33 should be included as an explanatory alongside all potential confounding variables such as parasite burdens, and the resulting fitted model estimate / z score for IL-33 be used to evaluate its importance.

We did not measure the expression of *Il33* by QPCR, only by RNASeq. Therefore, we have responded to Reviewer 3’s concerns re. the potential confounding effect of infection (particularly helminth infection) within our DGE analysis. As explained in more detail below, we have accounted for cestode burden (cestodes being the most prominent helminth infection in terms of biomass) in an extra DGE analysis. The addition of cestode burden had little effect on our results, with *Il33* still emerging as a top-responding gene in males.

Reviewer #2 (Recommendations for the authors):In general, I would recommend you tone down the claims about sex-specific effects; actually, the results presented here are of interest regardless of whether they are sex-specific or not. Given that such effects are at best marginally significant, I think this is something that is better addressed in the Discussion, rather than claimed in the title.

We agree with the Reviewer that our results are interesting regardless of whether they are sex-specific or not. As already mentioned above, we have tempered our claims about sex-specific effects.

Reviewer #3 (Recommendations for the authors):– Please revise and expand reporting of the immune gene expression data (Il-33 is a type 2 cytokine, not normally considered pro-inflammatory). Much of the discussion of inflammatory vs anti-inflammatory requires more nuance – there are, at a minimum, antibacterial (IL-17), antihelminthic (IL-33, GATA3), and anti-inflammatory (SOC3) pathways being flagged up, and it would also be important to test antiviral responses (interferon-stimulated genes).

Please see our more detailed response above re. classification of genes. We now mention in the results that *Il33* is involved in the antihelminthic response , and *Il17a* is involved in the antibacterial response. No interferon-stimulated genes were flagged up in our analyses so these (and antiviral responses more broadly) are not mentioned.

– Covariates detailed in Table 1 lack infection variables. These would be essential adjustments because the hypothesis is that all else being equal (i.e., at identical infection burdens, etc.), GC carriers should be more pro-inflammatory (males) or anti-inflammatory (females).

We understand the Reviewer’s concerns about our DGE analysis. As mentioned above in more detail, we have run an extra DGE analysis including cestode burden as a covariate (a parasite variable of relevance to IgE, as pointed out by the Reviewer). We found very similar results in this analysis, suggesting that the differences in immune gene expression that we see are not driven by cestode burden. We have presented the result of this extra DGE analysis as extra supplementary tables (see Appendix 1—table 4 and 5). We have also included the following caveat in our discussion:

“Some of the differences in immune phenotype that we observed may also be driven by difference in parasite infection (although we accounted for cestode burden in a follow-up analysis, we cannot rule this out).”

– The differential expression of Il33 (and any other genes/pathways of interest) should be followed up with statistical analysis to determine whether it is driven by worm infection, reproductive status, or other potential confounders in addition to / rather than haplotype. A GLMM containing covariates detailed in Table 1 + infection data would seem appropriate.

We did not measure the expression of *Il33* by QPCR, only by RNASeq. Therefore we have responded to these comments within our DGE analysis. As mentioned above, we have accounted for cestode burden in an extra DGE analysis. The addition of cestode burden had little effect on the results.

– Likewise, the effect of GC on male (and female?) reproductive success should be adjusted for the same potential confounders, like infection level.

As mentioned above, we have run extra GLMs for both females and males which include two parasite variables: proportion of samples taken from an individual that tested positive for *Babesia* and proportion of samples taken from an individual that tested positive for *Bartonella*. We found no difference in the main results – males with the GC haplotype still have fewer offspring, suggesting that infection is not acting as a confounder. This additional analysis is mentioned in the text: “We ran the same models including microparasite variables, but this had little effect on our results (see Appendix 1—table 14 and 15), suggesting that these effects were not driven by differences in microparasite infection”, The model output is also included in the supplementary information, see Appendix 1—table 14 and 15. For the majority of those individuals with known reproductive success, we did not have data on macroparasite burden (which requires culling). However, given the lack of association between macroparasite burden and GC haplotype in cross-sectional animals, and the lack of effect of including cestodes in the DGE analysis, we don’t believe including macroparasite variables would change these results.

– Figure 1A is quite uninformative. Please consider representing genes and top enriched pathways in a manner that allows readers to interpret the data.

As mentioned above, barcode plots such as this are commonly used representation of GSEA results (see e.g. Sadik *et al.* 2020, *Cell* 182, 1252–1270; Kumarasingha *et al.* 2020, *Frontiers in Immunology* 11, 582358). We would expect most readers to be familiar with them. As such, we would like to keep Figure 1A as it is. In order to aid the Reviewer and those readers who are unfamiliar with barcode plots, we have included some more information in the legend:

“Unstimulated immune gene expression: Barcode plots showing enrichment of the GO terms GO:0050729 (pro-inflammatory genes) and GO:0050728 (anti-inflammatory genes) in unstimulated splenocytes taken from individuals with (hGC+) vs. without (hGC-) the haplotype, showing that males with the haplotype have a pro-inflammatory bias, whereas females have an anti-inflammatory bias. In each plot, x-axis shows log fold change (logFC) in hGC+ vs. hGC-, black bars represent genes annotated with the GO terms and the worm shows relative enrichment”.

[Editors’ note: further revisions were suggested prior to acceptance, as described below.]

The manuscript has been improved but there are some remaining issues that need to be addressed.(1) Reviewer 2 has pointed out that it is difficult to understand how Fcer1a could be linked to IL17a production. This finding should be backed in the manuscript by some reference to previous evidence, or it should be made clear that this result is unexpected and difficult to explain. Reviewer 2 also suggests removing the result, which is another option.

We have added more justification on this point, evidenced with nine new references (please see below for details).

(2) Please see other suggestions by Reviewer 2 for improving clarity.

We have made all of Reviewer 2’s suggested changes to improve clarity (please see below for details).

Reviewer #2 (Recommendations for the authors):The authors have dealt with most of my comments in a satisfactory way, but there are still a few significant problems. In particular, the rationale for the experiment where T cells were stimulated ex vivo is highly unclear (see below for details).

We are glad that we have satisfactorily addressed most of Reviewer 2’s comments.

The conclusions are now better matched by data, although there are still quite strong claims about sex-specific effects even though there are few formal tests of this (i.e. tests for sex-by-genotype effects). More problematic is that the justification (L 578-581) provided for the analysis of ex vivo stimulated T cells is very vague. The problem with this assay is as follows: I believe that in splenocytes, Fcer1a is mainly (only?) expressed on basophils and eosinophils; thus, an effect of a Fcer1a polymorphism needs to involve these cells in some way. In your assay you stimulated T cells (which are abundant in splenocytes) in a non-specific way. Of the 18 cytokines measured, you find an effect of Fcer1a polymorphism on expression of IL17a, which is produced by T cells (but not by eosinophils/basophils). The problem is that I just cannot see how the Fcer1a polymorphism (which should have an effect in basophils/eosinophils, but not in T cells) could be linked to IL17a production by T cells; this would need to involve signalling from activated T cells to basophils/eosinophils, and then Fcer1a polymorphism-dependent signalling from basophils/eosinophils back to T-cells to affect IL17a production; I guess it is not impossible, but I am unaware of anything like that. I think you need to provide a sound and specific rationale for this assay; just saying "it could be acting indirectly on T-cells through various pathways, including via cytokine signalling, following Fcer1a expression by other cells" (L578-581) is not enough.

We have added more justification, evidenced with nine new references, on this point, around the line numbers indicated. The reviewer is correct that expression of the Fcer1a protein in the high-affinity IgE receptor is mainly associated with cells like basophils or eosinophils – although such expression is sometimes reported, or considered to be important, in other cells. However, it should be kept in mind that the cells assayed here were derived ex vivo from a wild organism and were also cultured in a complex global mixture – within which the T-cells that were specifically stimulated would have initially occurred alongside other cell types bearing functional high-affinity IgE receptor. Thus, the stimulated T-cells would previously (in vivo) and during the assay (ex vivo) have been under the influence of high-affinity IgE receptor-bearing cells. Given the modern attitude to interaction networks between cells, and the known cytokine-producing and immunoregulatory properties of high-affinity IgE receptor-bearing cells – including demonstrated effects on helper T-cells in ex vivo co-culture and in in vivo models (see the new references) – it does not seem surprising, or unreasonable to expect, that there might be indirect interactions with T-cells. In the event, we did, in fact, find evidence of an effect of *Fcer1a* polymorphism on the responses in T-cell-stimulated cultures that corroborated our independent findings – and this provides its own justification.

The presentation of the DGE analysis and GSEA of unstimulated splenocytes is improved, but still a bit confusing (L 199-238): (i) I think it would be good to make clear in the Results that the DGE analysis and GSEA are based on comparing individuals with the GC haplotype vs all others (so that the reader does not have to go the MandM to find this), (ii) using the term "top-responding" (L 206) implies that you are measuring the transcriptional response to some treatment, while you are just comparing baseline expression of GC vs others in males and females (or have I misunderstood something?); use "differentially expressed" or something similar instead, (iii) similarly, using the term "upregulated" (L 231) implies you measured the transcriptional response to some treatment; use "more highly expressed in individuals with GC…" or similar.

In response to point (i): We have now clarified in the Results that we compared individuals with vs. without the GC haplotype in the DGE analysis, by adding the following text:

“Differential gene expression (DGE) analysis was performed on unstimulated splenocytes taken from 53 males and 31 females assayed by RNASeq, with the aim of identifying individual genes which were differentially expressed between those individuals with and without a copy of the GC haplotype.”

We have done the same for the GSEA by editing/adding the following text in the Results:

“This analysis showed that both gene sets were more highly expressed in individuals with the GC haplotype than individuals without the haplotype, and that this was true for both males and females.”

In response to points (ii) and (iii): Throughout, we have replaced “top-responding” with “top differentially expressed”, and “upregulated” with “more highly relatively expressed in individuals with the GC haplotype” (or similar), as suggested by the Reviewer – adding the word ‘relatively’ to be precise (we report relative expression values, as detailed in the Materials and Methods).

When it comes to the ex vivo stimulation of T cells, I strongly suggest you remove these results (L240-254); they just don't make sense from an immunological point of view. Unfortunately, this is one of few analyses that actually shows a sex-by-genotype effects, so removing this would weaken the support for sex-specific effects of the Fcer1a polymorphism even further.

This point has been addressed above – and more justification has been provided in the methods.